# Beware of Calibration Data for Pruning Large Language Models

**Yixin Ji**[1,2], **Yang Xiang**[1,2], **Juntao Li**[1,2][*] **Qingrong Xia**[3],
**Ping Li**[3], **Xinyu Duan**[3], **Zhefeng Wang**[3], **Min Zhang**[1,2]
[1]School of Computer Science and Technology, Soochow University
[2]Key Laboratory of Data Intelligence and Advanced Computing, Soochow University
[3]Huawei Cloud, China
{jiyixin169,baldwin021129}@gmail.com;
{ljt,minzhang}@suda.edu.cn
{xiaqingrong,liping61,duanxinyu,wangzhefeng}@huawei.com;

## Abstract

As large language models (LLMs) are widely applied across various fields, model compression has become increasingly crucial for reducing costs and improving inference efficiency. Post-training pruning is a promising method that does not require resource-intensive iterative training and only needs a small amount of calibration data to assess the importance of parameters. Recent research has enhanced post-training pruning from different aspects but few of them systematically explore the effects of calibration data, and it is unclear if there exist better calibration data construction strategies. We fill this blank and surprisingly observe that calibration data is also crucial to post-training pruning, especially for high sparsity. Through controlled experiments on important influence factors of calibration data, including the pruning settings, the amount of data, and its similarity with pre-training data, we observe that a small size of data is adequate, and more similar data to its pre-training stage can yield better performance. As pre-training data is usually inaccessible for advanced LLMs, we further provide a self-generating calibration data synthesis strategy to construct feasible calibration data. Experimental results on recent strong open-source LLMs (e.g., DCLM, and LLaMA-3) show that the proposed strategy can enhance the performance of strong pruning methods (e.g., Wanda, DSnoT, OWL) by a large margin (up to 2.68%)[1].

## 1 Introduction

Recently, Large Language Models (LLMs) have exhibited remarkable performance and enormous potential in Natural Language Processing (NLP) and Artificial Intelligence (AI) (OpenAI, 2022; 2023; Bubeck et al., 2023; Yang et al., 2023). The success of LLMs is closely tied to scaling laws (Kaplan et al., 2020; Hoffmann et al., 2022): training language models with more parameters, using more data and greater computational resources leads to more powerful capabilities. However, LLMs with more parameters increase the difficulty and cost of deployment and inference. Therefore, much work has been devoted to compressing LLMs to achieve a trade-off between efficiency and performance, such as pruning (Ma et al., 2023; Xia et al., 2024) and quantization (Frantar et al., 2023; Huang et al., 2024; Shao et al., 2024).

Pruning is a model compression technique that has evolved over many years (LeCun et al., 1989) and remains full of potential and challenges. Based on the over-parameterization of neural networks, it aims to remove redundant parameters while minimizing the degradation of model performance. Pruning has been successfully applied to compress small to medium-sized neural networks. Through sparse training (Lee et al., 2019; Frankle & Carbin, 2019; Yuan et al., 2021; Lasby et al., 2024) or pruning-aware training (Sanh et al., 2020; Lagunas et al., 2021; Jiang et al., 2023) methods, it can achieve performance comparable to dense models with a high sparsity ratio ($\geq$70%). However, these

---

[*] Corresponding author.
[1]Code is available at https://github.com/Dereck0602/calibration_data

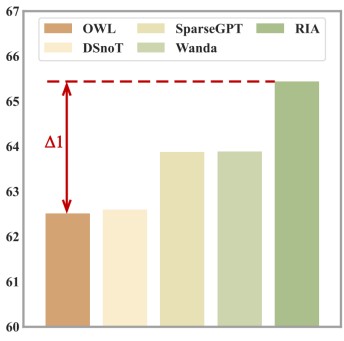
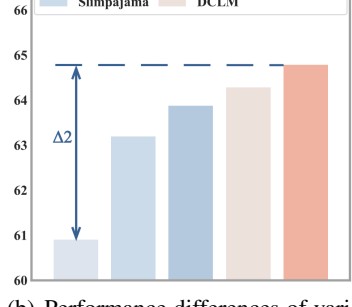
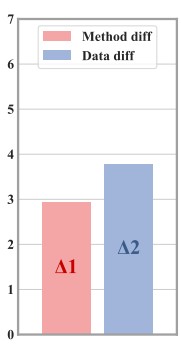

(a) Peformance differences of representative pruning methods with the commonly-used C4 calibration data.

(b) Performance differences of various calibration data on SparseGPT.

(c) Method differences v.s. data differences.

Figure 1: The effects of pruning methods and calibration data on commonsense reasoning tasks.

methods require iterative training, which is costly and time-consuming for LLMs with billions of parameters. As a result, post-training pruning that does not require iterative training has become the preferred approach for pruning LLMs.

The challenge of post-training pruning is how to perform training-free parameter importance estimation. Frantar & Alistarh (2023) note that simple parameter magnitude-based metrics perform poorly in post-training pruning with over 20% sparsity. Therefore, they use a small amount of calibration data to compute the inverse Hessian matrix, estimating parameter importance through second-order gradient information. Sun et al. (2024) propose a simpler method by using the product of weight magnitudes and the L2 norm of the corresponding input activations. Dong et al. (2024) utilize the genetic algorithm to search for the optimal combination of information from magnitude, activation, and gradient as an importance metric. Overall, current advanced parameter importance metrics rely on calibration data. Although most papers claim their pruning methods are robust to calibration data, Williams & Aletras (2024)'s empirical study challenges this view. They demonstrate the performance differences of various methods using different calibration data. Our experiments further revealed that the performance gains from selecting better calibration data can even surpass those of advanced pruning methods (Figure 1).

To learn more about calibration data, we design experiments to explore (1) the impact of calibration data with increased sparsity and varied pruning types, (2) the influence of the amount of calibration data, and (3) the selection strategy of calibration data. Our empirical results demonstrate that as sparsity increases, the performance differences among different calibration data become more pronounced, and simply increasing the data volume does not reduce this disparity. We further find that calibration data similar to the pretraining data yields better performance. Based on this, we propose the self-generation strategy to construct appropriate calibration data for pruning in practical settings with unavailable training data. To evaluate the effectiveness of our proposed calibration data sampling method, we conduct experiments on DCLM, LLaMA-2, and LLaMA-3 models. The results show that our proposed method performs better than the commonly used calibration data and is compatible with strong pruning methods by substantially improving their performance.

## 2 BACKGROUND

Model compression is a crucial way to improve inference efficiency by reducing the required memory, including pruning (Guo et al., 2023; Zhang et al., 2024b; Xia et al., 2024), quantization (Xiao et al., 2023; Lin et al., 2024), low-rank decomposition (Kaushal et al., 2023; Yuan et al., 2024; Wang et al., 2024; Ji et al., 2024), etc. The enormous memory requirements and inefficient inference speeds for LLMs urgently necessitate model compression. However, many successful model compression methods have required substantial computational resources for retraining, which limits their application for LLMs in low-resource settings. Therefore, post-training compression, which does not require retraining, has become a current research focus.

Post-training compression methods typically approximate model compression as an optimization problem for layer-wise compression (Frantar & Alistarh, 2022):

$$\min_{\hat{\boldsymbol{W}}_l}||\boldsymbol{W}_l\boldsymbol{X}_l - \hat{\boldsymbol{W}}_l\boldsymbol{X}_l||_F, \tag{1}$$

where $\boldsymbol{W}_l$, $\hat{\boldsymbol{W}}_l$ are the original and compressed $l$-th linear layer, respectively, and $\boldsymbol{X}_l$ is the input feature activation. For post-training pruning, to optimize the objective, OBC (Frantar & Alistarh, 2022) and SparseGPT (Frantar & Alistarh, 2023) utilize second-order gradient information to measure parameter importance and propose an efficient algorithm for computing the inverse Hessian matrix. Wanda (Sun et al., 2024) evaluates weight importance by combining their magnitudes with input activations without requiring backpropagation. Zhang et al. (2024c) propose the relative importance and activation metric (RIA), which integrates weight, input, and output activation. They also utilize the channel permutation to minimize pruning loss under N:M semi-structured pruning. Pruner-Zero (Dong et al., 2024) designs a genetic algorithm-based framework to automatically search the best pruning metric. Recently, several studies (Sung et al., 2024; Xu et al., 2024a; Yin et al., 2024) indicate that layer-wise compression, which typically applies a uniform sparsity rate across all layers and evaluates weight importance within the layer, often results in suboptimal performance due to the lack of overall consideration. Specifically, Xu et al. (2024a) proposes a differentiable pruning framework designed to search for optimal pruning rates for each layer. OWL (Yin et al., 2024) introduces outlier weighed layerwise sparsity, which relates the sparsity of each layer to the observed outliers in a proportional manner.

In the aforementioned post-training compression methods, calibration data is an indispensable component. Calibration data is a small subset randomly sampled from unlabeled pretraining text. Many methods (Frantar & Alistarh, 2023; Sun et al., 2024; Dettmers et al., 2024) claim their robustness to the quantity and distribution of calibration data, requiring only dozens or hundreds of samples with 2,048 sequence length. However, this conclusion is based on the perplexity of certain datasets (such as Wikitext2), which does not fully reflect the true capabilities of the LLMs. Even if perplexity shows no significant change, the compressed model may still experience substantial performance declines in downstream tasks (Jaiswal et al., 2024). Khanal & Capone (2024) suggest that using task-specific calibration data helps improve performance on specific downstream tasks. Williams & Aletras (2024) observe in extensive experiments that the selection of calibration data in post-training pruning and quantization methods significantly impacts downstream tasks' performance, especially post-training pruning, which is highly sensitive to calibration data. Shin et al. (2024) notice that the reconstruction error objective (Eq. 1) leads to overfitting on calibration data, and that self-generated calibration data can effectively mitigate the overfitting. Nevertheless, current research on calibration data remains under-explored, with few studies providing guidelines for selecting calibration data. Different from previous works, our paper (1) explores the impact of calibration data under varying sparsity ratios and types, (2) investigates the effect of data amount on various calibration data, not limited to the widely used C4 calibration data, (3) further addresses which calibration data is suitable for LLM pruning and provides a practical and effective method.

## 3 THE IMPACT OF CALIBRATION DATA FOR PRUNING

Though Williams & Aletras (2024) have noted that calibration data significantly impacts post-training pruning, there exist many open questions. How much does calibration data affect pruning performance? How does the amount of calibration data affect compressed model performance? What data sources are more suitable for calibration? We investigate these questions in this section.

### 3.1 EXPERIMENTAL DETAILS

**Dense Model** To study the impact of data from different sources on post-training pruning methods, we need a comprehensive knowledge of the data used in model training. We select the powerful and fully open-source LLM (including training data), DCLM-7B[2] (Li et al., 2024), as the dense model and conduct post-training pruning with different calibration data on it.

---

[2]https://huggingface.co/apple/DCLM-7B

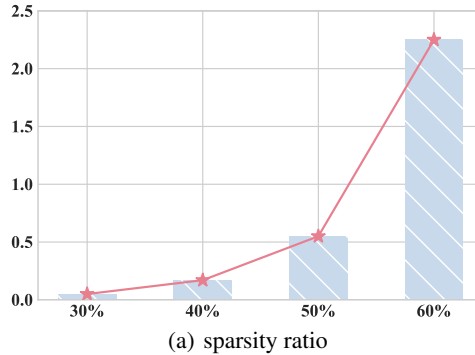
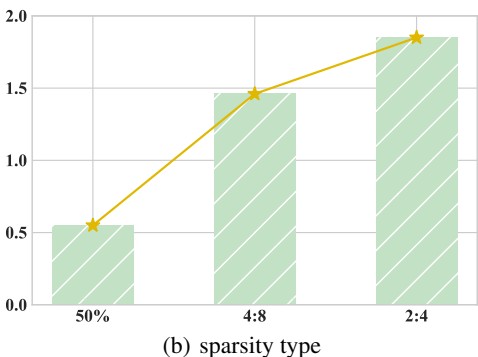

(a) sparsity ratio        (b) sparsity type

Figure 2: Pruning performance range ($Max.$-$Min.$) of different datasets (C4, Wikipedia, Slimpajama, DCLM) under various sparsity ratios (a) and sparsity types (b) on Wanda.

**Post-training Pruning Methods** We choose three competitive and representative post-training pruning methods for evaluation: Wanda (Sun et al., 2024), DSnoT (Zhang et al., 2024d) and OWL (Yin et al., 2024). These methods apply to both unstructured and semi-structured pruning.

**Calibration Data** We consider various data sources to be calibration data. Following the mainstream works, the calibration data sources are all from the unlabeled pre-trained corpus:

- C4 (Raffel et al., 2020)[3] is a widely used calibration data source, consisting of a large amount of multilingual web text filtered from Common Crawl. We sample from the English training set.
- Wikipedia[4] is a source of high-quality encyclopedic text. We use the first shard of the cleaned English version until `2023-11-01`.
- Slimpajama[5] is a cleaned and deduplicated version of RedPajama. It is a high-quality pre-training corpus with diverse sources, including C4, ArXiv, GitHub, Books, etc.
- DCLM (Li et al., 2024) is the pre-training data of DCLM-7B model. It includes 2.6T tokens extracted from Common Crawl. We sample from a subset[6] of the DCLM.

Aside from the experiments in Section 3.3, we follow prior works and randomly sample 128 sequences with 2048 tokens as calibration data. To mitigate the impact of sampling randomness, all our experiments repeat the calibration data sampling 20 times with different random seeds and report the average performance.

**Evaluation Tasks** Some pruning works focus on the perplexity of certain datasets while neglecting performance on various downstream tasks, which often fails to fully reflect the capabilities of compressed models. Therefore, we choose multiple widely used and challenging commonsense reasoning tasks for evaluation, including BoolQ (Clark et al., 2019), Winogrande (Sakaguchi et al., 2021), PIQA (Bisk et al., 2020), Hellaswag (Zellers et al., 2019), ARC-e, ARC-c (Clark et al., 2018) and MMLU (Hendrycks et al., 2021). For MMLU, we use a 5-shot setting, while all other tasks are evaluated in a zero-shot setting. Our evaluation code is based on the `lm-evaluation-harness` repository[7]. We report the average performance of these seven tasks.

### 3.2 HOW MUCH DOES CALIBRATION DATA AFFECT PRUNING PERFORMANCE?

In practical applications, evaluating and comparing the impact of different calibration data on pruned models inevitably consumes time and computational resources. Therefore, we wonder how significant the impact of calibration data is on pruning performance and whether it's worth our effort to seek optimal calibration data in research and practice. We consider different sparsity ratios and

---

[3]https://huggingface.co/datasets/allenai/c4

[4]https://huggingface.co/datasets/wikimedia/wikipedia

[5]https://huggingface.co/datasets/DKYoon/SlimPajama-6B

[6]https://huggingface.co/datasets/robbiegwaldd/dclm-micro

[7]https://github.com/EleutherAI/lm-evaluation-harness

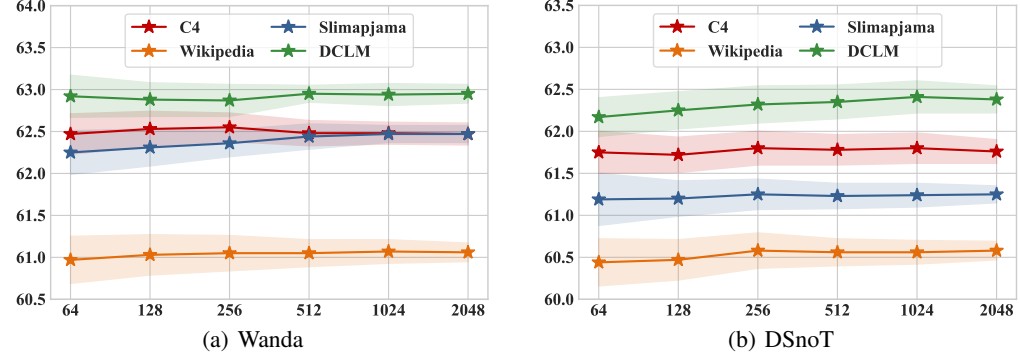

Figure 3: The impact of calibration data amount for different pre-training data resources (i.e., C4, Wikipedia, Slimapjama, DCLM) and pruning methods, i.e., Wanda (a) and DSnoT (b). Shaded areas represent the standard deviations of 20 random seeds.

sparsity types. Our experiments cover sparsity ratios ranging from 30% to 60%, and at 50% sparsity ratio, we further compare unstructured, 4:8 semi-structured, and 2:4 semi-structured sparsity types.

We use Wanda as an example to illustrate the model's performance range, defined as the difference between the maximum and minimum values, after pruning with four calibration data sets, as shown in Figure 2. More details on the performance of the different calibration data can be found in Figure 6 in Appendix A. Specifically, at low sparsity ratios (<50%), the performance difference between different calibration data is minimal, less than 0.1%. As sparsity increases, the impact of calibration data on pruning gradually amplifies, rising from a 0.5% difference at 50% sparsity to 2.3% at 60% sparsity. Notably, as shown in Figure 6, inappropriate calibration data can even have a negative effect at moderate sparsity levels. For instance, at 60% sparsity, using Wikipedia and Slimapjama as calibration data performs worse than magnitude pruning without any calibration data. For sparsity types, we observe that as the sparsity pattern becomes more structured, the choice of calibration data becomes increasingly important, with the maximum difference reaching 1.5% to 1.8%. We also report results on DSnoT and OWL in Appendix A. Although different pruning methods exhibit varying performance, they show similar trends regarding the impact of calibration data. **Overall, at moderate to high sparsity ratios and with semi-structured sparsity types, different calibration data significantly affect the performance of pruned LLMs**. For all pruning methods, higher sparsity ratios and more structured sparsity types are key to achieving effective inference acceleration. Therefore, paying more attention to the choice of calibration data is crucial.

### 3.3 IS CALIBRATION DATA FROM DIFFERENT SOURCES EQUALLY ROBUST TO DATA AMOUNT?

Currently, almost all post-training pruning methods for LLMs have empirically demonstrated robustness in terms of the amount of calibration data they use. Typically, model performance reaches a plateau when the data amount reaches 128, and more calibration data do not lead to additional performance gains. We wonder whether these methods are equally robust to the amount of data for calibration data from different sources. Can certain calibration data that lead to poorer pruned models be improved by increasing the data amount?

We perform Wanda and DSnoT pruning on DCLM-7B in the 2:4 semi-structured pruning setting. We randomly sample 64, 128, 256, 512, 1024, and 2048 samples from different data sources as calibration data. Figure 3 shows how the performance of pruned models changes with increasing data amount using different calibration data. We observe that **the average performance of pruned models is robust to data amount, regardless of the calibration data source**, with fluctuations of only 0.1%-0.2%. Therefore, we cannot expect that increasing the amount of calibration data will narrow the performance gap between different calibration data. Additionally, as the data amount increases, the standard deviation of the pruned model's performance decreases.

## 3.4 WHAT CALIBRATION DATA IS SUITABLE FOR PRUNING?

Since the choice of calibration data is crucial and cannot be improved by increasing the amount alone, we have to figure out what calibration data is more suitable for pruning. We propose two reasonable hypotheses: (1) The more similar the calibration data is to the training data of the LLMs, the better the pruning performance. (2) The higher the quality of the calibration data, the better the pruning performance.

To verify the hypotheses, we perform three post-training pruning methods on DCLM-7B with various calibration data in the 2:4 semi-structured pruning setting. We report our results in Table 1. Among these data, using DCLM from the training data as calibration data consistently achieves the best performance. C4 and Slimpajama, which are also extracted from Common Crawl,

Table 1: Pruning performance of three pruning methods with four different sources of calibration data.

| Method | C4 | Wikipedia | Slimpajama | DCLM |
|--------|-----|-----------|------------|------|
| Wanda | $62.52_{0.21}$ | $61.03_{0.21}$ | $62.31_{0.22}$ | $\mathbf{62.88}_{0.20}$ |
| DSnoT | $61.71_{0.21}$ | $60.48_{0.24}$ | $61.20_{0.21}$ | $\mathbf{62.25}_{0.22}$ |
| OWL | $63.40_{0.19}$ | $62.23_{0.19}$ | $63.10_{0.22}$ | $\mathbf{63.60}_{0.16}$ |

perform slightly worse. In contrast, the source of Wikipedia differs significantly from the other three datasets. Although Wikipedia is recognized as high-quality data, it shows the worst performance, falling short of DCLM by 1.3% to 1.8%. Therefore, we assert that the quality of calibration data is not the primary factor affecting pruning performance. We further quantify the similarity between different calibration data and the training data. We utilize the MinHash-LSH algorithm to encode the 3-grams of C4, SlimPajama, Wikipedia, and DCLM, calculating their Jaccard similarities. The results show that the Jaccard similarity between C4 and DCLM is 0.070, SlimPajama is 0.016, and Wikipedia is 0.008. This indicates that C4 is the most similar to the training data, followed by SlimPajama, while Wikipedia has the lowest similarity. This ranking aligns with their performance as calibration data in pruning. Therefore, we believe that **the similarity of calibration data to the training data has a more significant impact on pruning performance than the quality of the calibration data. Training data or data similar to the training data is better suited as calibration data**. We conjecture that this may be due to LLMs learning the patterns in the training data better. Therefore, using data with similar patterns as calibration data during the pruning process can more accurately reflect the importance of model parameters.

## 4 CALIBRATION DATA SAMPLING METHOD

In the Section 3, our empirical study of the open-source DCLM-7B model demonstrates that selecting calibration data similar to the training data can yield better pruning performance. However, in practical scenarios, the training data of many LLMs is not publicly available to users. In this section, we will propose the "self-generating then sampling" strategy for sampling calibration data when the training data is unavailable. Formally, given a dataset $\mathcal{D}$ as the source of calibration data and an LLM $\mathcal{M}$ pre-trained on an inaccessible dataset $\mathcal{D}_t$, we aim to sample $n$ instances from $\mathcal{D}$ as calibration data $\mathcal{D}_c$ that has a similar distribution to $\mathcal{D}_t$.

Recently, Xu et al. (2024b) disclosed that LLMs internalize patterns such as language structure, word distribution, and even commonsense knowledge from the training data during the training process. Due to their auto-regressive nature, LLMs leverage these internalized patterns when predicting the next token, producing the generated text similar to the training data. Thus, we propose using self-generated synthetic data as a proxy for the training data for calibration in post-training pruning. Specifically, for a sample from the source of calibration data $\mathcal{D}$, we truncate the first $t$ tokens as the prefix and then allow the LLM $\mathcal{M}$ to generate contextually relevant subsequent content:

$$x_i \sim p_{\mathcal{M}}(x_{<i}), i = t \cdots N. \tag{2}$$

After generating the data, we filter the synthetic data to prevent low-quality generated data from negatively impacting pruning effectiveness. We calculate each generated sample's perplexity and filter the $k\%$ samples with the highest perplexity. Higher perplexity indicates that the patterns are not well-fitted by the LLM and may differ significantly from the training data, making them unsuitable as calibration data.

## 5 EXPERIMENTS

### 5.1 EXPERIMENTAL DETAILS

To evaluate the effectiveness of our proposed calibration data sampling method, we apply it to various LLMs, including DCLM-7B, LLaMA-2-7B, LLaMA-2-13B (Touvron et al., 2023) and LLaMA-3-8B (Dubey et al., 2024). As described in Section 3.1, we use C4, Wikipedia, Slimpajama, and DCLM as baselines for calibration data, employing three post-training pruning methods: Wanda, DSnoT, and OWL, to prune the dense models. In the main experiments, we report performance at the 60% sparsity ratio. We follow previous work to evaluate the compressed LLMs' language modeling and commonsense reasoning capabilities. We do not use the Wikitext2 dataset, which is common in most papers for evaluating language modeling ability, as its similarity to Wikipedia may introduce bias when assessing the impact of different calibration data on language modeling ability. Instead, we choose the Alpaca (Taori et al., 2023) dataset, distinct from all four calibration data sources, as our language modeling test data.

When replicating DSnoT and OWL, we follow the hyperparameter settings detailed in their papers. During the self-generation process, we use Top-$k$ and Top-$p$ sampling to improve the diversity of the generated data. Specifically, we set the $p$-value to 0.95, the $k$-value to 50, and the temperature to 0.6. We apply the repetition penalty of 1.2 to avoid repeatedly generating low-quality fragments. We randomly sample 5,000 examples from C4, Slimpajama, Wikipedia, and DCLM respectively for generation. In the filtering phase, we eliminate the top 20% of samples based on their perplexity.

### 5.2 OVERALL PERFORMANCE

We report the main results in Table 2 and Table 5. Overall, our self-generated synthetic calibration data exceeds other baseline calibration data in language modeling and commonsense reasoning tasks and is compatible with different pruning methods. On DCLM-7B, Wikipedia, which is not part of the pretraining data, achieves the greatest performance improvement through self-generating synthetic data. It improves performance in commonsense reasoning tasks by an average of 2.2% to 2.6% compared to the original Wikipedia data, and even surpasses the commonly used C4 calibration data, achieving an average increase of 0.8% to 1.2%. For C4 and Slimpajama, which partially overlap with the pretraining data, the self-generation strategy also yields a 0.9-1.5% improvement. On LLaMA family models, the self-generated synthetic data also performs better than the original data, with improvements ranging from approximately 0.9% to 1.1%, and surpasses the C4 data by about 0.3% to 0.5%. Surprisingly, the performance of the self-generated calibration data even exceeds that of calibration data sampled from the DCLM-7B training set, with an average improvement of 0.3% to 0.7%. We think this may be due to certain patterns in the calibration data that LLMs have not adequately learned. Using these patterns as calibration data may misestimate the importance of parameters. In contrast, due to the nature of maximum likelihood training, self-generated calibration data typically generates patterns that LLMs have better learned, thus avoiding using underrepresented patterns as calibration data. Additionally, we observe that regardless of the source of synthetic data, the pruned models' performances are similar. It indicates that self-generated calibration data is versatile, as it can generate suitable calibration data even when the available data is significantly different from the pretraining data.

## 6 DISCUSSION

### 6.1 IS THE SYNTHETIC CALIBRATION DATA SUITABLE FOR OTHER PRUNING SETTINGS?

We further validate the effectiveness of self-generated synthetic calibration data across more pruning settings. Table 3 illustrates the commonsense reasoning performance of DCLM-7B during Wanda pruning using different calibration data at unstructured 50% and 65% sparsity ratios, as well as semi-structured 4:8 and 2:4 settings. In all pruning settings, our synthetic calibration data either matches or exceeds the performance of the optimal calibration data from the training set

Table 3: Pruning performance of different calibration data.

| Setting | C4 | Wiki | Slim | DCLM | Syn |
|---------|-------|-------|-------|---------|---------|
| 50% | 69.43 | 69.07 | 69.26 | 69.62 | **69.64** |
| 65% | 57.22 | 53.97 | 56.10 | **58.14** | 58.11 |
| 4:8 | 66.27 | 64.82 | 66.17 | 66.28 | **67.02** |
| 2:4 | 62.52 | 61.03 | 62.31 | 62.88 | **63.61** |

Table 2: Pruning performance of different calibration data on DCLM-7B in 60% sparsity ratio. The best performance method is indicated in **bold**. Wiki, Slim, and Syn are abbreviations for Wikipedia, SlimPajama, and our synthetic data, respectively. Underline means the improved performance of synthetic calibration data over the original calibration data for a certain task. Δ denotes the average performance change of pruned models on commonsense reasoning tasks. ✓, ✗ and ✗̷ indicate that the calibration data belongs, does not belong, or partially belongs to DCLM-7B's pretraining data, respectively.

| Data | Pretrain | Alpaca (↓) | BoolQ | Winogrande | PIQA | Hellaswag | ARC-e | ARC-c | MMLU | Avg. | Δ |
|---|---|---|---|---|---|---|---|---|---|---|---|
| | | | | | *Wanda* | | | | | | |
| Wiki | ✗ | 9.99 | 72.05 | 68.40 | 74.33 | 64.79 | 73.14 | 39.91 | 42.20 | 62.12 | |
| w/ Syn | | **9.40** | 78.73 | 70.06 | 75.78 | 66.16 | 74.34 | 42.83 | 45.04 | 64.71 | +2.59 |
| C4 | ✗̷ | 9.67 | 78.47 | 70.27 | 75.12 | 66.32 | 72.84 | 40.84 | 43.31 | 63.88 | |
| w/ Syn | | 9.57 | 78.81 | 70.52 | 75.95 | 66.35 | 74.23 | 42.01 | 45.64 | **64.78** | +0.90 |
| Slim | ✗̷ | 9.76 | 78.56 | 70.16 | 74.27 | 65.07 | 72.37 | 39.94 | 43.40 | 63.40 | |
| w/ Syn | | 9.58 | 78.51 | 70.02 | 75.63 | 65.90 | 74.12 | 42.13 | 45.26 | 64.51 | +1.11 |
| DCLM | ✓ | 9.54 | 79.11 | 70.51 | 75.13 | 66.25 | 73.37 | 41.66 | 44.58 | 64.37 | |
| w/ Syn | | 9.59 | 79.23 | 70.69 | 75.64 | 66.17 | 74.04 | 42.01 | 45.42 | 64.74 | +0.37 |
| | | | | | *DSnoT* | | | | | | |
| Wiki | ✗ | 10.16 | 69.97 | 68.08 | 73.95 | 63.23 | 72.09 | 38.69 | 41.63 | 61.09 | |
| w/ Syn | | **9.40** | 77.58 | 69.20 | 75.38 | 64.76 | 73.27 | 41.66 | 44.53 | **63.77** | +2.68 |
| C4 | ✗̷ | 9.81 | 76.11 | 69.44 | 74.76 | 65.08 | 72.10 | 39.08 | 41.62 | 62.60 | |
| w/ Syn | | 9.56 | 75.61 | 69.30 | 75.56 | 65.13 | 73.06 | 41.11 | 45.24 | 63.57 | +0.97 |
| Slim | ✗̷ | 9.87 | 75.58 | 69.21 | 73.80 | 63.88 | 71.37 | 38.63 | 42.25 | 62.10 | |
| w/ Syn | | 9.62 | 76.08 | 69.27 | 75.09 | 64.57 | 73.16 | 40.97 | 44.57 | 63.39 | +1.29 |
| DCLM | ✓ | 9.70 | 77.39 | 69.36 | 74.63 | 64.89 | 72.06 | 39.83 | 43.73 | 63.13 | |
| w/ Syn | | 9.52 | 76.56 | 68.35 | 75.55 | 64.70 | 73.43 | 41.43 | 44.81 | 63.55 | +0.42 |
| | | | | | *OWL* | | | | | | |
| Wiki | ✗ | 9.96 | 75.27 | 67.11 | 74.25 | 63.07 | 73.01 | 38.35 | 38.75 | 61.40 | |
| w/ Syn | | **9.20** | 78.45 | 68.92 | 76.03 | 65.18 | 73.72 | 40.29 | 42.73 | 63.61 | +2.21 |
| C4 | ✗̷ | 9.52 | 78.14 | 68.90 | 75.55 | 65.22 | 72.46 | 38.24 | 39.04 | 62.51 | |
| w/ Syn | | 9.31 | 78.55 | 68.67 | 76.38 | 65.45 | 74.05 | 40.03 | 42.94 | **63.72** | +1.21 |
| Slim | ✗̷ | 9.59 | 78.09 | 68.69 | 74.56 | 64.00 | 72.35 | 37.95 | 39.84 | 62.21 | |
| w/ Syn | | 9.32 | 78.56 | 68.71 | 75.83 | 64.47 | 73.81 | 40.44 | 43.61 | 63.64 | +1.43 |
| DCLM | ✓ | 9.38 | 78.45 | 69.47 | 75.10 | 65.07 | 72.76 | 38.81 | 40.73 | 62.91 | |
| w/ Syn | | 9.28 | 78.80 | 67.77 | 75.90 | 64.77 | 73.84 | 40.56 | 43.67 | 63.61 | +0.70 |

DCLM. Notably, the synthetic data improve performance by approximately 0.8% in the two semi-structured pruning settings. Since semi-structured pruning can achieve practical inference acceleration and advanced GPUs already support 2:4 sparse tensor cores. Thus, we think the self-generated synthetic calibration data will effectively enhance the performance of pruned models in real-world deployment.

## 6.2    HOW DOES PREFIX LENGTH AFFECT THE PERFORMANCE OF SYNTHETIC DATA?

The prefix length during self-generation is a crucial hyperparameter. If the prefix is too short, the synthetic text is likely to be far from the semantics of the original text; if it is too long, the synthetic calibration data may retain excessive patterns from the original text. Therefore, it is essential to explore the selection of prefix length. Our experiments range from 0 to 1024 prefix lengths, where a prefix length of 0 indicates only a special token representing the start of the text. Figure 4 shows the trend of commonsense reasoning performance as the prefix length varies. Once there is a prefix, the performance exceeds that of the original calibration data. However, longer prefixes do not yield better results, as performance gradually declines with increased prefix length. The results indicate that using 1 to 4 tokens as a pre-

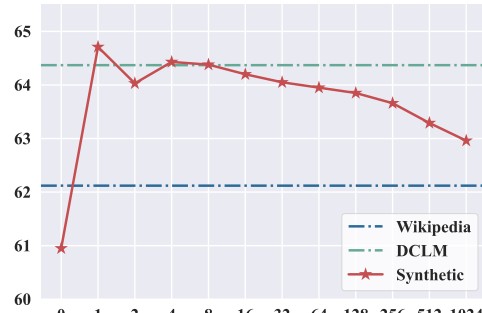

Figure 4: Wanda pruning performance using self-generated synthetic calibration data with different prefix lengths.

fix is optimal. This suggests that semantic consistency
with the original text is not critical in synthetic calibration data; instead, the key is to avoid retaining
patterns that could have negative effects.

### 6.3 HOW DOES PERPLEXITY-BASED DATA FILTERING AFFECT PRUNING PERFORMANCE?

After generating synthetic data, we employ a simple perplexity-based method to filter low-quality data. Is this perplexity-based filtering method effective, and what should the filtering rate be? We conduct experiments on the DCLM-7B model. As shown in Table 4, even without any filtering strategy, the synthetic data outperforms the original data. The perplexity-based filtering has proved to be a simple yet effective approach, with the best pruning performance at a filtering rate of 10%-20%. As the filtering rate increases, pruning effectiveness gradually declines, ultimately matching the performance of the unfiltered data. Therefore, we recommend filtering only the outliers based on perplexity, as overly aggressive filtering may compromise the diversity of the calibration data, negatively impacting pruning performance.

Table 4: Impact of perplexity-based data filtering.

| Data | Alpaca ($\downarrow$) | Commonsense |
|---|---|---|
| Wiki | 9.99 | 62.12 |
| w/o filter | - | 64.49 |
| 10% filter | 9.42 | 64.76 |
| 20% filter | 9.40 | 64.71 |
| 30% filter | 9.40 | 64.49 |
| 40% filter | 9.47 | 64.51 |

### 6.4 WHETHER SELF-GENERATED SYNTHETIC CALIBRATION DATA IS MORE SIMILAR TO TRAINING DATA?

In Section 3.4, we assert that data similar to the training data is more suitable as calibration data for post-training pruning. Based on the auto-regressive generation characteristics of LLMs, we propose using self-generated data as an approximation of the training data. But is the self-generated synthetic data truly similar to the model's training data than other calibration data? We use an efficient and effective Min-K%++ method (Zhang et al., 2024a) for measuring. Min-K%++ notes that after maximum likelihood training, the probability distribution of the training data always lies at local maxima along the input dimensions. Therefore, for a given token sequence $(x_{<t}, x_t)$, if the sequence is belong to the training data, the $p(x_{<t}, x_t)$ should be higher than that of other candidate tokens in the vocabulary. The Min-K%++ is formulated as follows:

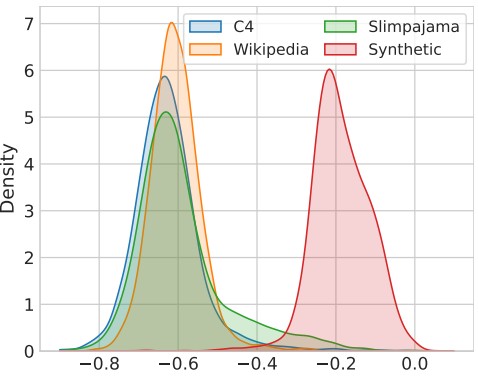

Figure 5: The Min-50%++ score distribution of C4, Wikipedia, Slimpajama and self-generated synthetic data.

$$W(x_{<t}, x_t) = \frac{logp(x_t|x_{<t}) - \mu_{x_{<t}}}{\sigma_{x_{<t}}},$$

$$\text{Min-K\%++}(x) = \frac{1}{|min\text{-}k\%|} \sum_{(x_{<t}, x_t) \in min\text{-}k\%} W(x_{<t}, x_t), \tag{3}$$

where $\mu_{x_{<t}}$, $\sigma_{x_{<t}}$ is the expectation and standard deviation of the next token's log probability given the prefix $x_{<t}$, respectively. $min\text{-}k\%$ refers to choosing the bottom $k\%$ of subsequences based on scores from the sequence $x$. Thus, the higher a sample's Min-K%++ score, the more likely it is to appear in the training data. Figure 5 uses kernel density estimation to show the distribution of Min-K%++ values for C4, Wikipedia, SlimPajama and our self-generated synthetic data. We can clearly observe that the self-generated synthetic data has higher Min-50%++ scores than the other calibration data. It indicates that the self-generated synthetic calibration data is indeed similar to the training data, confirming the validity of using self-generated data as a proxy for the training data.

## 7 CONCLUSION AND FUTURE WORK

In this paper, we highlight the critical role that calibration data plays in post-training pruning for LLMs. Through systematic exploration, we demonstrate that calibration data similar to the original training data leads to superior pruning performance. To address the challenge of inaccessible

training data in practical scenarios, we propose a self-generating synthetic calibration data strategy, which effectively samples suitable calibration data for LLMs. Experimental results on the DCLM, LLaMA-2, and LLaMA-3 models demonstrate that our method significantly outperforms existing common-used calibration data. We firmly believe that calibration data, as an essential part of post-training pruning, still holds significant potential for further research.

Our work still has some limitations that are worth exploring further. First, we do not fully optimize the hyperparameters when generating synthetic calibration data, such as using more advanced decoding strategies or refined filtering methods. We believe that improving these details could further enhance the effectiveness of the synthetic calibration data. Second, our experiments are limited to unstructured and semi-structured pruning on 7B-13B LLMs. In future work, we will validate our method on 70B LLMs and in structured pruning scenarios. Additionally, we will continue to explore how to synthesize high-quality instruction data as calibration data to help compress aligned LLMs.

## ACKNOWLEDGMENTS

We want to thank all the anonymous reviewers for their valuable comments. This work was supported by the National Science Foundation of China (NSFC No. 62206194), the Natural Science Foundation of Jiangsu Province, China (Grant No. BK20220488), the Young Elite Scientists Sponsorship Program by CAST (2023QNRC001), and the Priority Academic Program Development of Jiangsu Higher Education Institutions.

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

# A  MORE STUDIES ON DIFFERENT SPARSITY

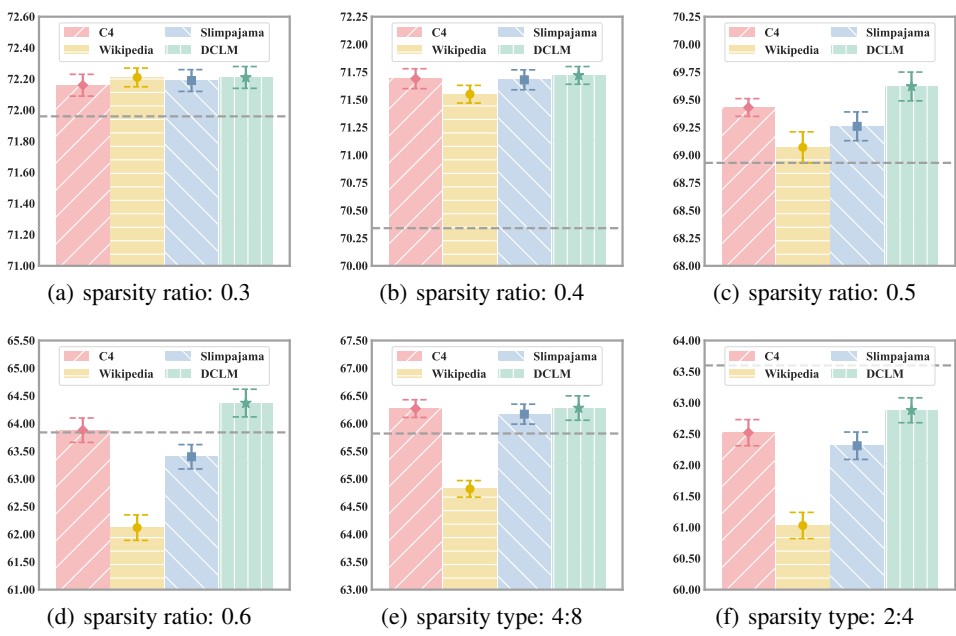

Figure 6: Pruning performance of different datasets (C4, Wikipedia, Slimpajama, DCLM) under various sparsity ratios (a-d) and sparsity types (e-f) on Wanda. The gray dash lines represent the performance of magnitude-based pruning.

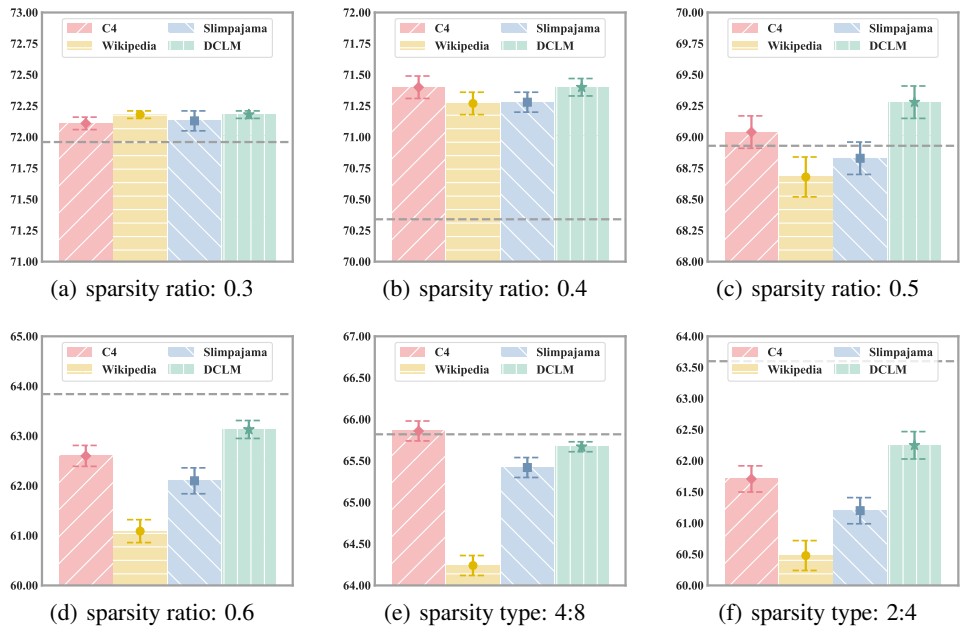

Figure 7: Pruning performance of different datasets (C4, Wikipedia, Slimpajama, DCLM) under various sparsity ratios (a-d) and sparsity types (e-f) on DSnoT. The gray dash lines represent the performance of magnitude-based pruning.

# B  MORE RESULTS OF SYNTHETIC CALIBRATION DATA

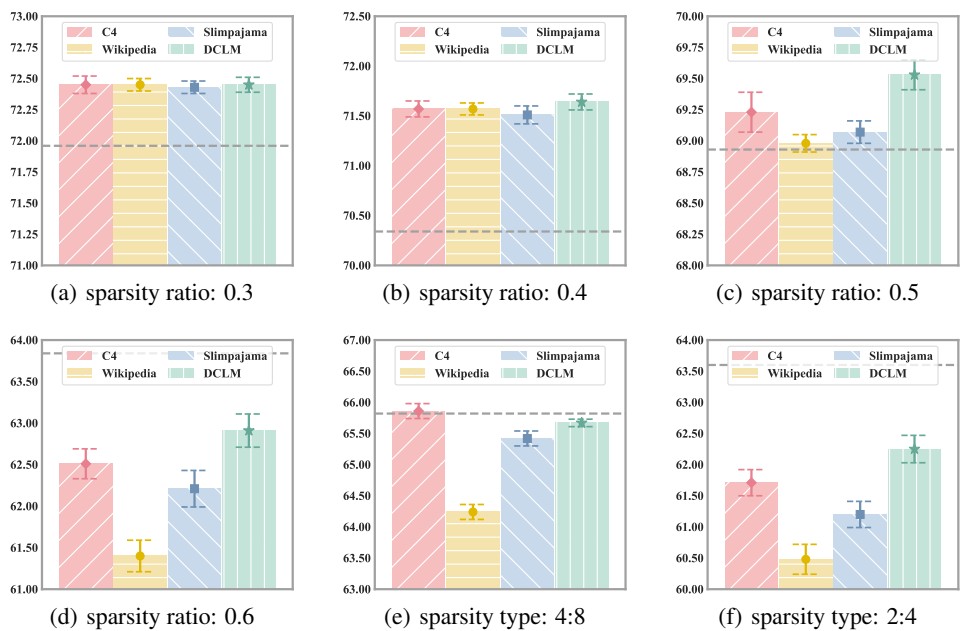

Figure 8: Pruning performance of different datasets (C4, Wikipedia, Slimpajama, DCLM) under various sparsity ratios (a-d) and sparsity types (e-f) on OWL. The gray dash lines represent the performance of magnitude-based pruning.

Table 5: Pruning performance of different calibration data on LLaMA-2-7B in 60% sparsity ratio. The best performance method is indicated in **bold**. Wiki, Slim, and Syn are abbreviations for Wikipedia, SlimPajama, and our synthetic data, respectively. Underline means the improved performance of synthetic calibration data over the original calibration data for a certain task. $\Delta$ denotes the average performance change of pruned models on commonsense reasoning tasks.

| Method | Data | Alpaca (↓) | BoolQ | Winogrande | PIQA | Hellaswag | ARC-e | ARC-c | MMLU | Avg. | $\Delta$ |
|---|---|---|---|---|---|---|---|---|---|---|---|
| Wanda | Wiki | 10.42 | 66.80 | 63.84 | 70.55 | 56.69 | 64.78 | 34.23 | 22.94 | 54.26 | |
| | w/ Syn | **9.62** | 68.29 | 64.40 | 71.49 | 58.89 | 64.73 | 35.41 | 24.01 | 55.32 | +1.06 |
| | C4 | 10.42 | 66.30 | 64.50 | 71.12 | 58.92 | 64.92 | 33.91 | 23.06 | 54.68 | |
| | w/ Syn | 10.07 | 67.46 | 64.15 | 71.38 | 59.05 | 65.64 | 33.83 | 23.92 | 55.06 | +0.38 |
| | Slim | 10.23 | 66.83 | 63.68 | 71.10 | 57.54 | 64.68 | 33.98 | 22.95 | 54.39 | |
| | w/ Syn | 9.92 | 67.91 | 64.63 | 71.45 | 58.52 | 65.28 | 33.93 | 23.29 | 55.00 | +0.61 |
| | DCLM | 9.88 | 68.92 | 64.25 | 71.15 | 58.72 | 64.81 | 33.98 | 23.65 | 55.07 | |
| | w/ Syn | 9.77 | 68.90 | 64.56 | 71.71 | 58.90 | 65.24 | 34.47 | 23.61 | **55.34** | +0.27 |
| DSnoT | Wiki | 10.92 | 66.24 | 62.72 | 70.55 | 55.55 | 64.10 | 33.16 | 23.05 | 53.62 | |
| | w/ Syn | 10.40 | 65.44 | 64.01 | 71.49 | 57.77 | 64.86 | 34.30 | 23.90 | 54.54 | +0.92 |
| | C4 | 10.88 | 65.25 | 64.04 | 71.22 | 57.15 | 64.40 | 32.82 | 23.45 | 54.05 | |
| | w/ Syn | 9.90 | 66.18 | 64.72 | 71.00 | 57.19 | 64.86 | 33.45 | 24.87 | **54.61** | +0.56 |
| | Slim | 10.76 | 65.66 | 63.66 | 70.82 | 56.17 | 64.43 | 32.51 | 23.15 | 53.77 | |
| | w/ Syn | 10.04 | 65.23 | 63.22 | 70.84 | 56.56 | 65.11 | 33.11 | 23.67 | 53.97 | +0.20 |
| | DCLM | 10.37 | 66.65 | 63.99 | 71.44 | 56.77 | 64.56 | 33.30 | 23.73 | 54.35 | |
| | w/ Syn | **9.82** | 66.24 | 64.01 | 71.00 | 57.64 | 64.86 | 33.70 | 24.09 | 54.51 | +0.16 |
| OWL | Wiki | 9.30 | 66.50 | 66.05 | 71.82 | 61.90 | 67.57 | 35.89 | 26.07 | 56.54 | |
| | w/ Syn | 9.13 | 69.85 | 66.38 | 73.18 | 62.86 | 67.89 | 35.07 | 26.34 | 57.37 | +0.83 |
| | C4 | 9.19 | 66.73 | 67.34 | 72.74 | 62.87 | 67.54 | 35.68 | 26.20 | 57.02 | |
| | w/ Syn | 9.11 | 68.47 | 67.40 | 72.52 | 62.99 | 66.75 | 35.41 | 27.28 | 57.26 | +0.24 |
| | Slim | 9.21 | 67.52 | 66.91 | 72.32 | 62.25 | 66.70 | 34.91 | 26.05 | 56.67 | |
| | w/ Syn | **9.04** | 69.30 | 67.80 | 72.31 | 62.56 | 67.17 | 35.75 | 26.86 | 57.39 | +0.72 |
| | DCLM | 9.08 | 69.79 | 67.94 | 72.39 | 62.73 | 67.06 | 35.85 | 26.45 | 57.46 | |
| | w/ Syn | 9.10 | 69.57 | 67.72 | 72.63 | 62.60 | 67.59 | 35.84 | 26.32 | **57.47** | +0.01 |

Table 6: Pruning performance of different calibration data on LLaMA-2-13B in 60% sparsity ratio. The best performance method is indicated in **bold**. Wiki, Slim, and Syn are abbreviations for Wikipedia, SlimPajama, and our synthetic data, respectively.

| Method | Data | Alpaca | BoolQ | Winogrande | PIQA | Hellaswag | ARC-e | ARC-c | MMLU | Avg. |
|---|---|---|---|---|---|---|---|---|---|---|
| Wanda | C4 | 8.99 | 77.36 | **68.68** | **75.45** | **66.51** | 69.18 | 39.74 | 26.80 | 60.53 |
| | Wiki | 9.21 | 74.39 | 67.97 | 74.97 | 64.39 | 68.66 | 38.62 | 24.96 | 59.14 |
| | Slim | 8.76 | 76.82 | 68.42 | 75.25 | 65.18 | 69.03 | 39.56 | 28.01 | 60.32 |
| | DCLM | **8.73** | **77.50** | 68.37 | 75.16 | 66.34 | 69.95 | 40.15 | 27.98 | 60.78 |
| | Syn | **8.73** | 77.06 | **68.68** | 75.19 | 66.25 | **70.03** | **40.19** | **29.06** | **60.92** |
| DSnoT | C4 | 9.03 | 77.16 | 66.60 | **74.92** | **65.76** | 69.81 | 38.45 | 25.73 | 59.77 |
| | Wiki | 9.34 | 76.02 | 65.89 | 74.43 | 63.84 | 68.93 | 37.95 | 25.19 | 58.89 |
| | Slim | 9.03 | 76.31 | 66.79 | 74.84 | 64.44 | 70.13 | 38.33 | 26.97 | 59.69 |
| | DCLM | 9.04 | **77.22** | 67.56 | 74.52 | 65.38 | 69.94 | 38.72 | 26.97 | 60.04 |
| | Syn | **8.96** | 77.09 | **67.64** | 74.54 | 65.33 | **70.29** | **39.68** | **27.08** | **60.23** |
| OWL | C4 | 7.56 | 78.92 | 70.02 | 75.95 | 69.12 | 70.90 | 41.14 | 32.75 | 62.69 |
| | Wiki | 8.25 | 77.93 | 69.47 | 75.23 | 68.13 | 71.20 | 39.23 | 31.75 | 61.85 |
| | Slim | 7.68 | 79.41 | 69.69 | 75.55 | 68.42 | 70.60 | 40.19 | 32.47 | 62.33 |
| | DCLM | **7.33** | 79.85 | **70.23** | 75.57 | **69.21** | **71.62** | 40.48 | **33.77** | **62.96** |
| | Syn | 7.35 | **79.05** | 69.61 | **76.50** | 69.11 | 71.51 | **41.55** | 31.19 | 62.65 |

Table 7: Pruning performance of different calibration data on LLaMA-3-8B in 60% sparsity ratio. The best performance method is indicated in **bold**. Wiki, Slim, and Syn are abbreviations for Wikipedia, SlimPajama, and our synthetic data, respectively.

| Data | BoolQ | Winogrande | PIQA | Hellaswag | ARC-e | ARC-c | MMLU | Avg. |
|---|---|---|---|---|---|---|---|---|
| C4 | 69.02 | 60.55 | 67.98 | 49.47 | 59.95 | 30.59 | **23.60** | 51.59 |
| Wiki | 66.82 | 59.02 | 67.40 | 47.14 | 59.79 | 29.67 | 24.14 | 50.57 |
| Slim | 66.86 | 60.11 | 67.53 | 48.07 | 59.38 | 29.96 | 23.52 | 50.77 |
| DCLM | **70.14** | 61.17 | 67.83 | 49.97 | **60.04** | 31.16 | 23.22 | 51.93 |
| Syn | 70.03 | **61.88** | **68.06** | **50.11** | 59.85 | **31.66** | 23.19 | **52.11** |

