# OpenReview forum: "Beware of Calibration Data for Pruning Large Language Models"
_ICLR.cc/2025/Conference — ICLR 2025 Poster_

### Official Review · Reviewer_4fLd · 2024-10-19

**Soundness:** 2
**Presentation:** 3
**Contribution:** 1
**Rating:** 3
**Confidence:** 4

**Summary:**

This paper investigates the impact of calibration data in the pruning of large language models (LLMs). This work mainly repeats some work that has been done by Williams & Altetras (EMNLP2024), which investigates the impact of calibration data in the pruning and quantization of large language models. The authors present evidence that the quality and type of calibration data can impact pruning performance, at times more so than advanced pruning methods themselves, reflecting the results done by Williams & Altetras (EMNLP2024). They propose a self-generating calibration data synthesis strategy to create effective calibration datasets when access to training data is limited.

**Strengths:**

1. **Originality:** In addition to the models included by Williams & Altetras (EMNLP2024), the authors also tested with DCLM-7B. This model is designed to showcase the effectiveness of systematic data curation. They propose a self-generating calibration data synthesis strategy.
2. **Quality:** The paper provides a systematic exploration, supported by experimental results demonstrating how different calibration datasets affect the performance of pruned models.
3. **Clarity:** The writing is reasonably clear and easy to follow. The objective is straightforward.
4. **Significance:** The findings have significant implications for practitioners in the field, although it has been highlighted by previous work already.

**Weaknesses:**

1. **Lack of Novel Contribution:** The study builds on important findings by Williams & Aletras (EMNLP2024), and the findings are already been proven and previous work has done further comparing quantization and pruning.
2. **Lack of downstream tasks experiments:** the authors only consider pruning performance and it does not necessarily reflect the downstream tasks. Previous work done by Williams & Altetras (EMNLP2024) has done a much more comprehensive evaluation of a wide range of downstream tasks.
3. **No explanation on pruning performance:** The paper primarily evaluates "pruning performance," but fails to provide a clear explanation of this metric. It's unclear whether this refers to pruning error, signal-to-noise ratio (SNR), or another measure. The authors neither explain their calculation method nor cite a source for this metric.
4. **Experimentation with Diverse Datasets:** The experiments predominantly focus on a narrow range of calibration datasets and models. Including a broader set of datasets could provide more generalizable results and strengthen the conclusions drawn about the effectiveness of their proposed methods.
5. **Validation or discussion of choices in methods:** There are some variables actually can be potentially impact the results, such as why 5000 samples from the Wikipedia data for generation, and why eliminate the top 20%.

**Questions:**

Could the authors provide further clarification on how efficient their proposed calibration data synthesis method is, e.g., what are the minimum data points it needs to generate for calibration?

---

> ### Author Response · Authors · 2024-11-23
>
> Thanks for your time and effort in reviewing our paper.  We wonder if there might be some misunderstandings about our work, and we hope our response will clarify these points.
>
> * **Lack of Novel Contribution**
>
> Thanks for your feedback. We try to look for Williams & Altetras (EMNLP2024) as you mentioned, but Williams & Altetras do not publish in EMNLP 2024. We speculate that you might be referring to the work of Williams & Aletras (ACL 2024)[1]. In the related work section (Lines 133-137), we briefly discussed the differences between Williams & Aletras' work. Here, we provide a more detailed explanation:
>
> (1) Williams & Aletras aimed to explore the impact of the source and amount of calibration data on LLM compression but did not provide specific guidance on how to select suitable calibration data. In contrast, our work further addresses which calibration data is suitable for LLM pruning and provides a practical and effective method.
>
> (2) Williams & Aletras first highlighted the importance of calibration data for LLM compression, particularly in post-training pruning methods. However, they did not quantify its impact. Through extensive experiments, we explore the impact of calibration data under varying sparsity ratios and types. Our results show that as the sparsity ratio increases and the sparsity type becomes more structural, the choice of calibration data becomes increasingly critical. Additionally, we compare several advanced pruning algorithms and find that the impact of calibration data surpasses the performance gains from improvements in pruning algorithms.
>
> (3) Williams & Aletras explore the impact of calibration data amount on model compression, but they were limited to the common-used C4 calibration data. In contrast, our research compares the performance of pruned models with calibration data from different sources as the data amount increases. We find that increasing the amount of calibration data from different sources does not improve pruning performance, indicating that the performance differences between different calibration datasets cannot be compensated for by simply increasing the data amount.
>
> If you are not referring to the aforementioned paper, we sincerely hope you can provide a more specific title so that we can address your concerns.
>
> * **Lack of downstream tasks experiments**
>
> In our experiments, we evaluate the performance of LLMs on various downstream tasks (see Lines 189-196), including BoolQ, Winograde, PIQA, Hellaswag, ARC-e, ARC-c, and MMLU.
> The table below lists the evaluation datasets used in 6 papers. Most works evaluate performance on 6-7 tasks. We use the six most commonly used datasets and additionally evaluate the MMLU dataset. MMLU alone includes 57 different tasks, so we think our evaluation on downstream tasks is also sufficiently solid.
> |                        | BoolQ | Winograde | PIQA | Hellaswag | ARC-e | ARC-c | OBQA | RTE | LAMBDA | Storycloze |
> |------------------------|-------|-----------|------|-----------|-------|-------|------|-----|--------|------------|
> | sparsegpt              |      |           | √    |           | √     | √     |    |     | √      | √          |
> | wanda                  | √     | √         |     | √         | √     | √     | √    | √   |        |            |
> | DSnoT                  |      |           | √    | √         | √     | √     | √    |     |        |       √     |
> | OWL                    | √     | √         |     | √         | √     | √     | √    | √   |        |            |
> | LLM-Pruner             | √     | √         | √    | √         | √     | √     | √    |  |      |          |
> | Williams & Aletras (2024 ACL) | √ | √     | √    | √         | √     | √     | √    | √   | √      | √          |
>
> To address your concerns, we report the results of Wanda for the remaining four tasks on DCLM-7B in the below table, showing that pruned models using pretraining data or synthetic data as calibration significantly outperform those using other calibration data.
>
> |                | LAMBDA  | OBQA  | RTE   | Storycloze | avg   |
> |----------------|---------|-------|-------|------------|-------|
> | c4             | 61.03   | 39.36 | 63.47 | 72.60      | 59.11 |
> | wikipedia      | 60.11   | 39.96 | 61.52 | 71.02      | 58.15 |
> | slimpajama     | 60.52   | 39.00 | 62.45 | 71.78      | 58.44 |
> | dclm           | 62.35   | 40.00 | 65.34 | 72.54      | 60.06 |
> | syn            | 59.56   | 39.80 | 68.59 | 72.40      | **60.09** |
>
> * **No explanation on pruning performance**
>
> We provide a detailed description of our pruning performance measure in lines 189-196. It measures the average accuracy of the pruning model on multiple commonsense reasoning tasks. SparseGPT, Wanda, DsnoT, OWL, and Williams & Aletras (2024 ACL) all evaluate pruning model performance based on commonsense reasoning tasks.
>
> [1] Miles Williams and Nikolaos Aletras. On the impact of calibration data in post-training quantization and pruning. 2024 ACL

---

> > ### Author Response · Authors · 2024-11-23
> >
> > * **Experimentation with Diverse Datasets**
> >
> > To strengthen our conclusions, we add two more calibration data sources: ArXiv and BookCorpus. The table below reports the performance of these two calibration datasets on DCLM-7B. It shows that using ArXiv and BookCorpus as calibration data results in post-pruning model performance similar to that of Wikipedia. In terms of data quality, both ArXiv and BookCorpus are high-quality pretraining datasets. However, in terms of similarity, ArXiv and BookCorpus, derived from scientific literature and books respectively, differ significantly from web data. This result further supports the conclusion presented in Section 3.4.
> >
> > |       | BoolQ | Winograde | PIQA | Hellaswage | ARC-e | ARC-c | MMLU | avg  |
> > |-------|-------|-----------|------|------------|-------|-------|------|------|
> > | **Wanda** |       |           |      |            |       |       |      |      |
> > | ArXiv | 77.81 | 67.86     | 74.02| 63.73       | 73.31 | 42.05 | 43.71| 63.21|
> > | BookCorpus  | 77.77 | 71.19     | 75.17| 65.95       | 69.44 | 38.99 | 43.92| 63.20|
> > | **DSnoT** |       |           |      |            |       |       |      |      |
> > | ArXiv | 73.03 | 67.40     | 73.56| 62.58       | 72.01 | 40.68 | 42.53| 61.69|
> > | BookCorpus  | 77.47 | 71.62     | 74.83| 65.19       | 58.91 | 38.26 | 42.50| 62.68|
> > | **OWL**   |       |           |      |            |       |       |      |      |
> > | ArXiv | 76.85 | 67.59     | 74.02| 62.37       | 72.74 | 39.51 | 40.70| 61.97|
> > | BookCorpus  | 75.90 | 70.37     | 75.16| 64.69       | 69.35 | 37.54 | 36.98| 61.43|
> >
> > * **Validation or discussion of choices in methods**
> >
> > Regarding the removal of the top 20% PPL samples, we have already discussed this in detail in Section 6.3, and we hope the ablation study can address your concerns. As for the amount of self-generated data, we intuitively select 5,000 samples. Since only 128 calibration data points are needed, generating more synthetic data will not improve performance and will waste computational resources. We also conduct experiments with fewer synthetic data points, and the results are as follows:
> >
> > |      | 500   | 1000  | 2000  | 3000  | 4000  | 5000  |
> > |------|-------|-------|-------|-------|-------|-------|
> > | Avg acc| $64.64_{0.14}$ | $64.86_{0.08}$ | $64.52_{0.24}$ | $64.72_{0.12}$ | $64.61_{0.08}$ | $64.71_{0.12}$ |
> >
> > The results in the table show that the amount of synthetic data does not affect the performance of the pruning model significantly. This result also indicates that our synthetic calibration data is efficient, requiring only 500-1,000 data points to generate for calibration.

---

> > > ### Comment · Reviewer_4fLd · 2024-11-26
> > >
> > > Thanks for your reply and sorry for the late feedback.
> > >
> > > Yes, Williams & Aletras (ACL 2024) is the one.
> > >
> > > For the three differences:
> > > - a sampling setup proposed ( **“self-generating then sampling”[1] has already been proposed**).
> > > - they did not quantify its impact,
> > > - Williams & Aletras limited to C4 calibration data, you have tried more.
> > >
> > > I acknowledge the contribution but it is marginal as it is mainly repeated in the previous experiments and the “self-generating then sampling” method has already been proposed.
> > >
> > > [1] Williams, M., Chrysostomou, G., & Aletras, N. (2024). Self-calibration for Language Model Quantization and Pruning. arXiv preprint arXiv:2410.17170.

---

> > > > ### Author Response · Authors · 2024-11-27
> > > > **Response to reviewer 4fLd**
> > > >
> > > > Thanks for your efforts as a reviewer; however, I kindly wish to express a differing perspective.
> > > >
> > > > 1. Our paper systematically investigated the impact of calibration data on the performance of pruned models and proposed a guideline for selecting calibration data. Through controlled experiments on the advanced LLM DCLM-7B with open-source pretraining data, we demonstrated that the similarity between calibration data and pretraining data, rather than the amount or quality of the data, is the key factor affecting the performance, which guided us to propose the self-generation method. In contrast, Williams et al. (2024.10, arXiv) did not delve into empirical research to explore why synthetic data is suitable for calibration.
> > > >
> > > > 2. "Self-calibration for Language Model Quantization and Pruning" was submitted to arXiv on October 22, 2024, while our paper was submitted to ICLR 2025 on October 1, 2024. **Williams et al. (2024.10 arxiv)'s paper was released more than 20 days after our submission for review. Therefore, we think papers published after ours should not be seen as diminishing the contributions of our work.** According to "ICLR 2025 Reviewer Guide", if a paper was published (i.e., at a peer-reviewed venue) on or after July 1, 2024, authors are not required to compare their own work to that paper.
> > > >
> > > > In summary, in terms of content, our paper is more foundational and concludes the empirical guideline for constructing suitable calibration data. In terms of timeline, our paper is released earlier than Williams et al. (2024.10, arXiv).

---

> > > > > ### Comment · Reviewer_4fLd · 2024-11-29
> > > > >
> > > > > Thank you for your clear explanation, I will increase my score while as stated before, I acknowledge the contribution but it is marginal due to similar works we discussed.

---

### Official Review · Reviewer_4EBf · 2024-10-28

**Soundness:** 4
**Presentation:** 3
**Contribution:** 3
**Rating:** 8
**Confidence:** 5

**Summary:**

This paper studies the impact of calibration data in the post-training pruning of LLMs, which shows that calibration data significantly affects pruned model performance as pruning difficulty increases, surpassing the improvements from advanced pruning algorithms. The authors also find that using training data or data similar to it as calibration data significantly boosts pruned model performance. Since pre-training data is often unavailable for advanced LLMs, the paper proposes a strategy for self-generating calibration data. Extensive experiments on multiple LLMs and pruning methods confirm the effectiveness of the proposed synthetic calibration data.

**Strengths:**

1. This paper introduces a criterion and construction strategy for choosing calibration data in post-training pruning, supported by extensive experimental validation.
2. The authors conduct experiments on various LLMs and pruning methods, with multiple repetitions, to eliminate the effects of randomness.
3. The paper is well-organized, clearly presenting the empirical studies, methodology, experiments, and results, making it easy for readers to follow the authors' arguments.

**Weaknesses:**

1. This paper only conducts experiments on unstructured and semi-structured pruning settings and does not validate the effectiveness of synthetic calibration data in more practical structured pruning.
2. The synthetic calibration data is not a method first proposed by the authors. A recent work by Shin et al.[1] also proposed synthetic calibration data. However, the authors do not discuss the differences between that work and the others.
3. This paper only uses data from Wikipedia to generate synthetic data. Why do you not validate the effectiveness of synthetic data generated from other sources?

[1] Shin, Sungbin, et al. "Rethinking Pruning Large Language Models: Benefits and Pitfalls of Reconstruction Error Minimization." arXiv preprint arXiv:2406.15524 (2024).

**Questions:**

1. Where does Figure 6 reflect the results of magnitude-based pruning?
2. Are the conclusions and method presented in this paper applicable to LLM quantization?

---

> ### Author Response · Authors · 2024-11-21
>
> Thanks for your careful review and insightful advice. The acknowledgment of our paper's contributions and the accurate summation of its highlights give us the confidence to continue improving it.
>
> * **Impact of synthetic calibration data on structured pruning**
>
> We evaluate the effectiveness of our synthetic calibration data on LLM-Pruner.  The results are as follows:
> | DCLM-7B      | BoolQ | Winograde | PiQA  | Hellaswag | ARC-e | ARC-c | MMLU  | Avg   |
> |--------------|-------|-----------|-------|-----------|-------|-------|-------|-------|
> | C4           | 62.57 | 60.10     | 77.49 | 68.36     | 70.94 | 40.74 | 34.59 | 59.26 |
> | Wikipedia    | 62.09 | 57.58     | 74.35 | 61.30     | 71.41 | 39.51 | 38.73 | 57.85 |
> | Slimpajama   | 61.96 | 59.00     | 75.37 | 64.23     | 70.03 | 38.08 | 34.61 | 57.61 |
> | DCLM         | 64.80 | 66.60     | 75.15 | 68.98     | 71.87 | 42.88 | 32.85 | 60.45 |
> | Syn          | 63.27 | 61.17     | 76.44 | 65.77     | 76.81 | 45.90 | 36.60 | **60.85** |
>
> Based on these results, we believe that synthetic calibration data can further improve the performance of structured pruning.
>
> * **The main differences between this work and Shin et al. (2024.6 arxiv)**
>
> Thank you for mentioning the work of Shin et al. We apologize for overlooking this important related work in our paper. We will include a citation and discussion of the differences in the revised version. Shin et al. primarily focused on optimizing the objective for post-training pruning and proposed various strategies to reduce reconstruction error. They observed that MSE objectives tend to overfit calibration data and suggested using self-generated calibration data to mitigate this issue. However, their evaluation was limited to improvements in reconstruction error and did not report performance on downstream tasks. In contrast, our work focuses on a comprehensive investigation of the impact of calibration data on pruned model performance, addressing key questions such as when calibration data matters, whether performance gaps between different calibration datasets can be narrowed, and how to construct appropriate calibration data. Additionally, the details of our synthetic calibration data generation method differ from theirs.
>
> * **The effectiveness of synthetic data generated from other sources**
>
> We use C4, Slimpajama, Wikipedia,DCLM to generate synthetic calibration data. The results are shown below:
> | Dataset       | BoolQ | Winograde | PiQA  | Hellaswag | ARC-e | ARC-c | MMLU  | Avg   | $\Delta$  |
> |---------------|-------|-----------|-------|-----------|-------|-------|-------|-------|--------|
> | C4            | 78.47 | 70.27     | 75.12 | 66.32     | 72.84 | 40.84 | 43.31 | 63.88 |        |
> | &ensp;&ensp;w/ syn        | 78.81 | 70.52     | 75.95 | 66.35     | 74.23 | 42.01 | 45.64 | 64.78 | +0.90  |
> | Wikipedia     | 72.05 | 68.40     | 74.33 | 64.79     | 73.14 | 39.91 | 42.20 | 62.12 |        |
> | &ensp;&ensp;w/ syn        | 78.73 | 70.06     | 75.78 | 66.16     | 74.34 | 42.83 | 45.04 | 64.71 | +2.59  |
> | Slimpajama    | 78.56 | 70.16     | 74.27 | 65.07     | 72.37 | 39.94 | 43.40 | 63.40 |        |
> | &ensp;&ensp;w/ syn        | 78.51 | 70.02     | 75.63 | 65.90     | 74.12 | 42.13 | 45.26 | 64.51 | +1.11  |
> | DCLM          | 79.11 | 70.51     | 75.13 | 66.25     | 73.37 | 41.66 | 44.58 | 64.37 |        |
> | &ensp;&ensp;w/ syn           | 79.23 | 70.69     | 75.64 | 66.17     | 74.04 | 42.01 | 45.42 | 64.74 | +0.37  |
>
> We find that regardless of the source of the synthetic data, the performance of the pruned model shows minimal differences and significantly outperforms the original data (except for DCLM, which is the part of the training data).
>
> * **Explain the Figure 6**
>
> We apologize for our negligence. The gray dashed lines in Figure 6, 7, 8 represent magnitude-based pruning.
>
> * **Apply to LLM quantization**
>
> Williams & Aletras[1] find that existing quantization methods are very robust to calibration data, so our paper focuses solely on LLM pruning. We reasonably speculate that synthetic calibration data will not lead to significant performance improvements for LLM quantization.
>
> [1] Miles Williams and Nikolaos Aletras. On the impact of calibration data in post-training quantization and pruning. In Lun-Wei Ku, Andre Martins, and Vivek Srikumar (eds.), Proceedings of the 62nd Annual Meeting of the Association for Computational Linguistics (Volume 1: Long Papers), pp. 10100–10118, Bangkok, Thailand, August 2024. Association for Computational Linguistics. URL https://aclanthology.org/2024.acl-long.544.

---

> ### Comment · Reviewer_4EBf · 2024-11-23
> **Thanks for the response.**
>
> Thanks for the response. After reading carefully, I am satified with the work and will keep the score.
>
> BTW, I can't entirely agree with reviewer 4fLd's point of overlap with EMNLP-2024, as no evidence has been provided to support this issue. In addition, the EMNLP-2024 conference has just been held, and the submission time of ICLR is much earlier than the EMNLP-2024 accepted notification. It is unclear what the Limited Novelty will be.
>
> There was no intention of offense. It was just my personal opinion.

---

> > ### Public Comment · ~Namhoon_Lee3 · 2025-03-11
> > **Shin et al. (2024)**
> >
> > Dear authors,
> >
> > Hope this email finds you well.
> >
> > We're the authors of the paper "Rethinking Pruning Large Language Models: Benefits and Pitfalls of Reconstruction Error Minimization" by Shin et al., published at EMNLP 2024.
> >
> > We have thoroughly enjoyed reading your work and appreciate its contributions. However, we noticed that our work, despite its relevance to yours---as acknowledged by both the reviewer and yourself---has not yet been discussed. We would therefore like to kindly request that it be appropriately addressed in your paper. If there is anything we can do to assist in this process, please let us know. Otherwise, we would greatly appreciate it if this could be incorporated into the camera-ready version as promised. Thank you in advance for your consideration.
> >
> > Best regards,
> > Namhoon Lee (on behalf of Shin et al.)

---

> > > ### Public Comment · ~Yixin_Ji2 · 2025-03-12
> > >
> > > Dear Prof. Lee,
> > >
> > > Thank you for your timely reminder. We've updated our camera-ready version to include a discussion of your EMNLP 2024 paper "Rethinking Pruning Large Language Models: Benefits and Pitfalls of Reconstruction Error Minimization". Appreciate your understanding and helpful alert.
> > >
> > > Best regards,
> > >
> > > Yixin

---

### Official Review · Reviewer_9hYo · 2024-11-03

**Soundness:** 3
**Presentation:** 2
**Contribution:** 3
**Rating:** 5
**Confidence:** 3

**Summary:**

Large language models with numerous parameters substantially increase deployment and inference complexity and costs. To mitigate this, post-training parameter pruning can be used which exploits the fact that neural networks are often over-parametrized. It operates selectively removing redundant parameters while aiming to preserve performance as measured using a sample of calibration data.
The key contributions of this paper are: (i) a (plausibly) novel data synthesis strategy for calibration data, and (ii) an investigation into the effects of size, quality, and distribution of calibration data, across different pruning hyperparameters.
Additionally, the paper examines major hyperparameter choices within their strategy and perform additional analysis to show that their synthesis method generates data that is distributed similar to the training data.

**Strengths:**

The paper productively expands on prior work to answer unanswered follow up questions related to the influence of calibration data on pruning and delivers insightful findings through a set of reliable experiments.
It proposes a novel and intuitive approach for the synthesis of calibration data and evaluates it empirically and theoretically while experimentally justifying major hyperparameter choices. They show that the approach can improve by up to 2.6% over using an out-of-distribution calibration dataset.
The paper also clearly describes background, relevant pruning approaches, the problem statement and proposed approach for calibration data synthesis as well as experimental results.

**Weaknesses:**

The main results are not so well represented. In Table 2, the proposed calibration data synthesis approach frequently falls behind other sources of calibration data. It’s not highlighted in the table (e.g., using colors or otherwise) whether each source was present in the training set of the evaluated LLM. That is, it makes sense to have separate comparisons for the proposed approach with each of (i), data the model was not trained on and (ii), data the model was trained on, but these seem to be mixed up in one table making it hard to interpret the quality of the results by looking at the table. The statement “Overall, our self-generated synthetic calibration data outperforms other baseline calibration data in language modeling and commonsense reasoning tasks and…” is not well justified because the remaining of the paragraph focuses on Wikipedia and C4 and its not obvious from the table that it outperforms all sources consistently over all tasks.

The paper involves some redundancies. For instance, the introduction as well as background seem to closely repeat the literature review. The questions are mentioned in the introduction then later again in section 3. Moreover, the choice of words in some of the sentences used is inadequate. For instance, the use of “value more” in “We fill this blank and surprisingly observe that the effects of calibration data even value more than designing advanced pruning strategies.” Take note as well that the paper does not convey that this “values more” than designing more advanced pruning strategies and that’s nontrivial to prove. Constructs such as “while different calibration data’s impact on pruning performance still lacks systematical exploration.” also make the abstract harder to read compared to if it was something like “while the impact of calibration data used has been…”.

**Questions:**

Suggestion (I): decompose or improve the table to highlight matching or exceeding the performance of using calibration data from the actual training set and exceeding the performance compared to calibration datasets belonging to other distributions.

Suggestion (II): avoid redundancy in repeating the literature review and possibly summarize the questions in the introduction. In the literature review, the name of the technique corresponding to each citation could be mentioned as well.
Suggestion (III): improve the abstract to better reflect the outcomes of the paper and be easier to read.
Suggestion (IV): Mention somewhere that the paper will first proceed by answering the calibration data related questions and then propose a new a novel technique for its generations. Typically, one expects the main novel contribution to come first.

---

> ### Author Response · Authors · 2024-11-24
>
> Thanks for your time and effort in providing valuable feedbacks, which is crucial in improving our work.
>
> * **Decompose or improve the table to highlight matching or exceeding the performance of using calibration data from the actual training set and exceeding the performance compared to calibration datasets belonging to other distributions.**
>
> Following the above suggestion, we have made the following modifications in the updated PDF file:
>
> 1. In Table 2, we clearly distinguish the relationship between calibration data and pretraining data. For DCLM-7B, DCLM is its open-source pretraining data, Wikipedia does not belong to pretraining data, while C4 and Slimpajama partially overlap with the pretraining data.
>
> 2. To clearly demonstrate the performance improvements of synthetic data, we add results using synthetic data generated from C4, Slimpajama, and DCLM. Regardless of the data source, models pruned with self-generated calibration data consistently surpass those using the original calibration data by up to 2.68% of averaged performance across different tasks.
>
> 3. Since we add more experiments with synthetic data, we move the results for LLaMA 2-7B to the Appendix to avoid Table 2 taking up too much space.
>
> 4. We revise the description of the experimental results in Section 5.2 based on the updated main results.
>
> * **Avoid redundancy in repeating the literature review and summarize the questions in the introduction.**
>
> 1. We remove repeated literature review either from introduction or background for conciseness.
>
> 2. We also improve the description of related work by mentioning the name of the technique corresponding to each citation.
>
> 3. Additionally, we have summarized the questions in the introduction.
>
> We summarize "However, many open questions regarding calibration data remain under-explored.  For example, how does the impact of calibration data change with increased sparsity and structure of pruning? Can increasing the amount of calibration data narrow the performance gap between various datasets? What type of data is suitable for calibration? How do we select the appropriate calibration data in practice?" as "To learn more about calibration data, we design experiments to explore (1) the impact of calibration data with increased sparsity and varied pruning types, (2) the influence of the amount of calibration data, and (3) the selection strategy of calibration data."
>
> * **Improve the presentation of abstract to better reflect the outcomes of the paper and be easier to read.**
>
> 1. We have improved sentences to avoid inadequate wording. We change "We fill this blank and surprisingly observe that the effects of calibration data even value more than designing advanced pruning strategies." to "We fill this blank and surprisingly observe that calibration data is also crucial to post-training pruning, especially for high sparsity. "
>
> 2. We have simplified complex sentences in the abstract. We replace "Previous research has primarily focused on designing advanced pruning methods, while different calibration data’s impact on pruning performance still lacks systematical exploration." with "Recent research has enhanced post-training pruning from different aspects but few of them systematically explore the effects of calibration data, and it is unclear if there exist better calibration data construction strategies."
>
> 3. We highlight the outcomes of our paper in the abstract. "Through controlled experiments on important influence factors of calibration data, including the pruning settings, the amount of data, and its similarity with pre-training data, we observe that a small size of data is adequate, and more similar data to its pre-training stage can yield better performance."
>
> * **Mention somewhere that the paper will first proceed by answering the calibration data related questions and then propose a new a novel technique for its generations.**
>
> Follow the suggestions, we have clearly sumamrize the main novel contributon of this work in Absratct and compare this works with the most relevant work in Background.
>
> Thank you once again for your detailed and valuable suggestions to improve the presentation of our paper. We have revised the paper with all changes highlighted in blue. We look forward to your inspiring response.

---

> > ### Author Response · Authors · 2024-11-30
> >
> > Dear Reviewer 9hYo,
> >
> > As the deadline is approaching, kindly let us know if you have any remaining concerns unanswered. We sincerely appreciate your valuable feedback and remain available to address any further concerns.
> >
> > Best regards, Authors

---

### Official Review · Reviewer_mY5n · 2024-11-04

**Soundness:** 3
**Presentation:** 3
**Contribution:** 2
**Rating:** 6
**Confidence:** 3

**Summary:**

This paper investigates the role of calibration data in post-training pruning for large language models (LLMs). The authors find that calibration data similar to the training data yields better performance when pruning LLMs for model compression. As many training datasets for LLMs are inaccessible, the authors propose a strategy to create synthetic calibration data, which outperforms commonly used datasets in experiments. This strategy involves generating synthetic text using the LLM and then filtering out low-quality data. This synthetic data is more similar to the training data and ultimately leads to better performance for pruned LLMs.

**Strengths:**

The paper effectively challenges the common assumption that post-training pruning methods are robust to the choice of calibration data. Recognizing the challenge of inaccessible training data, the paper introduces a "self-generating then sampling" strategy for constructing suitable calibration data. The paper provides a detailed examination of various aspects related to the self-generating calibration data strategy

**Weaknesses:**

While the paper shows a correlation between training data similarity and pruning performance, it doesn't explain why this connection exists. The paper's evaluation primarily centers on overall model performance. Investigating how calibration data affects the pruning of individual model components like attention heads or specific layers could be beneficial. This granular analysis would offer a more complete picture of how calibration data impacts different parts of the LLM.

**Questions:**

- What are the main differences between this work and the work by [1]?

- The authors say that "We can clearly observe that the self-generated synthetic data has higher Min-50%++ scores than the other calibration data. It indicates that the self-generated synthetic calibration data is indeed similar to the training data, confirming the validity of
using self-generated data as a proxy for the training data.". The conclusion is not entirely clear to me, can you explain how to conclude that synthetic calibration data is similar to the training data in this figure?

- While the paper aims to enhance general capabilities, the impact of using domain-specific calibration data for pruning models intended for specialized tasks remains unclear. do the authors have any intuition for that?

[1] Miles Williams and Nikolaos Aletras. On the impact of calibration data in post-training quantization
and pruning. In Lun-Wei Ku, Andre Martins, and Vivek Srikumar (eds.), Proceedings of the
62nd Annual Meeting of the Association for Computational Linguistics (Volume 1: Long Papers),
pp. 10100–10118, Bangkok, Thailand, August 2024. Association for Computational Linguistics.
URL https://aclanthology.org/2024.acl-long.544.

---

> ### Author Response · Authors · 2024-11-22
>
> Thanks a lot for your valuable comments. We hope the following responses can address your concern:
>
> * **Explain the correlation between training data similarity and pruning performance**
>
> We briefly analyze this issue in lines 379-383 of the paper. To address your concerns, we provide a more detailed explanation here. Based on the optimization objective of post-training pruning:
>
> $\underset{\hat{\boldsymbol{W}}_l}{\text{min}} \left\\| \boldsymbol{W}_l \boldsymbol{X}_l - \hat{\boldsymbol{W}}_l \boldsymbol{X}_l \right\\|_F,$
>
> calibration data affects $X_l$ in the formula. If the calibration data is not representative, the $X_l$ may not adequately reflect the LLM's broad general capabilities. This underrepresentation can lead to biased measurements of the importance of $W_l$ parameters during pruning, affecting performance. Through LLM pretraining, the training data is typically well represented by the LLM and captures most of its general abilities. Therefore, we believe using training data or data similar to it as calibration data defines a better pruning objective, enabling a more accurate estimation of LLM parameter importance while preserving the LLM's general capabilities.
>
> * **The fine-grained impact of calibration data on different modules.**
>
> Thanks for your valuable suggestion, which is highly insightful for understanding the impact of calibration data on pruning.
> Based on your suggestion, we conduct preliminary experiments to explore the impact of calibration data on sparsity ratio allocation across layers. We take OWL as an example and report the layer-wise pruning ratio allocation under the 60% sparsity ratio in the table below:
> |               | 2      | 4      | 6      | 8      | 10     | 12     | 14     | 16     |
> |---------------|--------|--------|--------|--------|--------|--------|--------|--------|
> | Wikipedia     | 56.84% | 60.62% | 61.44% | 63.46% | 64.20% | 64.39% | 64.00% | 63.57% |
> | Syn           | 56.16% | 60.03% | 60.74% | 62.73% | 63.48% | 63.74% | 63.20% | 63.06% |
> | |**18**           | **20**     | **22**     | **24**     | **26**     | **28**     | **30**     | **32**     |        |
> | Wikipedia     | 62.83% | 59.13% | 57.53% | 58.61% | 57.72% | 57.05% | 57.24% | 48.45% |
> | Syn           | 62.33% | 59.63% | 58.38% | 59.91% | 59.11% | 58.12% | 58.08% | 47.74% |
>
> Comparing the two allocation results, we find that when using synthetic data as calibration data, OWL automatically assigns higher pruning ratios to the middle layers (layers 20–30) while assigning lower ratios to other layers. This is consistent with the conclusion in ShortGPT[1] that middle-to-late layers are less important, while the layers at both ends are more critical. Thus, we think suitable calibration data not only improves the accuracy of parameter importance estimation but also optimizes layer-wise importance estimation, enabling more reasonable pruning ratio allocation.
>
> As for more fine-grained studies, such as specific attention heads, frankly, due to our limited understanding of how different attention heads affect model performance, we are currently unable to draw unified conclusions based on pruning results from different calibration data. Therefore, we have to leave this insightful topic for future research.
>
> * **Explain the Figure 5**
>
> Min-K%++ is a metric for pretraining data detection, where higher values indicate greater similarity to pretraining data and a higher likelihood of being part of it [2]. Figure 5 uses kernel density estimation to show the distribution of Min-K%++ values for C4, SlimPajama, Wikipedia, and self-generated synthetic data. The distribution curve for synthetic data is noticeably right-skewed, indicating that its Min-K%++ values are generally higher than the other three calibration data.
>
> [1] Men, Xin, et al. "Shortgpt: Layers in large language models are more redundant than you expect." arXiv preprint arXiv:2403.03853 (2024).
>
> [2] Zhang, Jingyang, et al. "Min-k%++: Improved baseline for detecting pre-training data from large language models." arXiv preprint arXiv:2404.02936 (2024).

---

> > ### Author Response · Authors · 2024-11-22
> >
> > * **The main differences between this work and Williams & Aletras (2024 ACL)**
> >
> > In the related work section (Lines 133-137), we briefly discussed the differences between Williams & Aletras' work. Here, we provide a more detailed explanation:
> >
> > (1) Williams & Aletras aimed to explore the impact of the source and amount of calibration data on LLM compression but did not provide specific guidance on how to select suitable calibration data. In contrast, our work further addresses which calibration data is suitable for LLM pruning and provides a practical and effective method.
> >
> > (2) Williams & Aletras first highlighted the importance of calibration data for LLM compression, particularly in post-training pruning methods. However, they did not quantify its impact. Through extensive experiments, we explore the impact of calibration data under varying sparsity ratios and types. Our results show that as the sparsity ratio increases and the sparsity type becomes more structural, the choice of calibration data becomes increasingly critical. Additionally, we compare several advanced pruning algorithms and find that the impact of calibration data surpasses the performance gains from improvements in pruning algorithms.
> >
> > (3) Williams & Aletras explore the impact of calibration data amount on model compression, but they were limited to the common-used C4 calibration data. In contrast, our research compares the performance of pruned models with calibration data from different sources as the data amount increases. We find that increasing the amount of calibration data from different sources does not improve pruning performance, indicating that the performance differences between different calibration datasets cannot be compensated for by simply increasing the data amount.
> >
> > * **The impact of using domain-specific calibration data for pruning models intended for specialized tasks**
> >
> > This question is very insightful for future research. We explore task-specific calibration data selection. We evaluate LLaMA-3.1-8b-instruct's performance on GSM8K and Math before and after compression (50% sparsity) with different calibration data. We try two methods for generating synthetic data. The first generates the CoT and answer based on the question, while the second generates both the question and the solution steps. The results are as follows:
> >
> > | Wanda                      | GSM8k | Math |
> > |----------------------------|-------|------|
> > | Dense                      | 78.2  | 37.7 |
> > | c4                         | 30.2  | 11.1 |
> > | gsm8k                      | 39.0  | 11.4 |
> > | syn (cot+answer)           | 39.0  | 14.3 |
> > | syn (question+cot+answer)  | 38.5  | 12.7 |
> >
> > Through this experiment, we think that for task-specific LLM compression, selecting data most relevant to the task (e.g., training data) as calibration data is a good choice. Compared to C4 calibration data, both synthetic data generation methods improve the pruned model's mathematical capabilities. Additionally, compared to directly using the GSM8K training set, synthetic data improves cross-dataset performance on similar tasks (e.g., MATH). Between the two methods, generating solution steps outperformed generating both questions and solutions. We think this is likely due to the varying quality of the generated questions, which may require further optimization of the question selection strategy.

---

> > > ### Comment · Reviewer_mY5n · 2024-11-23
> > > **TY**
> > >
> > > Thanks for the response. I will keep my score.

---

### Meta-Review · Area_Chair_Fn47 · 2024-12-12

**Metareview:**

Post-hoc pruning methods are a common and efficient approach for compressing pretraining language models; however, these approaches rely on some auxiliary data, or "calibration data," to estimate parameter importance. This work tests how the choice of calibration data affects the performance of post-hoc pruning methods on various downstream tasks. They find that the choice of data does affect performance: data similar to the original training data performs better, and this effect becomes stronger at higher rates of sparsity in the pruned model. The work then proposes a method for generating synthetic calibration data from the model directly (for the common case where an LLM's training data is not freely available), which is also effective for model pruning.

Strengths:
- This work provides several useful and actionable insights into how calibration data affect post-hoc pruning methods, and in my opinion after reading both papers, significantly builds on the insights in [1].
- The paper follows up on the presented analysis of calibration data to propose a general method for obtaining effective calibration data from the model, even when the underlying training distribution isn't known.
- The paper presents extensive experiments to support the findings in the paper across multiple models, data sources, and tasks.

Weaknesses:
- The paper doesn't explain why data that is more similar to the training distribution works better, though they present a hypothesis and reasoning as to why this occurs in the paper. Experiments testing this and other analyses—such as the results on domain-specific calibration data—would flesh out the findings significantly.
- The paper is, at some points, confusing and redundant. The authors should ensure that the next version of the paper clears up any lingering points of confusion and redundancies raised by the reviewers.

The reviewers bring up other weaknesses—the presentation of the main results (9hYo), description of the paper's novelty and contributions (mY5n, 9hYo), and synthetic data source ablation (4EBf), etc.—that are addressed in the author response and the revised paper.

[1] Miles Williams and Nikolaos Aletras. On the impact of calibration data in post-training quantization and pruning. In ACL 2024.

**Additional Comments On Reviewer Discussion:**

The authors provided detailed responses and numerous revisions to address the reviewer's concerns. Three of the four reviews responded to the rebuttal, and each chose to keep their scores.

---

### Decision · Program_Chairs · 2025-01-22

Accept (Poster)